# Assessing the Sustainability of China's Basic Pension Funding for Urban and Rural Residents

**Lanying Sun [1], Changhao Su [1,]\* and Xinghui Xian [2]**

[1]  College of Management and Economics, Tianjin University, Tianjin 300072, China; sunlytju@163.com
[2]  Petro China Southwest Pipeline Company, Chengdu 610041, China; xianxinghui@petrochina.com.cn
\*  Correspondence: such@tju.edu.cn

**Abstract:** This paper uses an actuarial model to evaluate the effect of poverty alleviation policy on the sustainability of China's basic pension funding for urban and rural residents (BPFUUR). Because of the poverty alleviation policy, China's basic endowment insurance for urban and rural residents (BEIURR) has been given the responsibility of helping the poor through insurance and by increasing the income of the elderly. On the basis of summarizing the current situation of BEIURR, this paper simulated and predicted the impact of poverty alleviation policy on the short term (2020–2025) and medium term (2025–2050) sustainability of China's BPFUUR. The results show that: (1) The sustainability of the fund will inevitably face challenges; (2) The per capita contribution level will be at a low level for a long time; (3) The birth tide brought by the two-child policy has a positive impact on the medium-term sustainability of the fund; (4) Poverty eradication mainly affects the short-term current deficit, but not the medium-term accumulated deficit; (5) The higher the payment, the better the sustainability of the funding in the short and medium term. According to the above results, this paper puts forward some countermeasures, such as insisting on family planning, optimizing the adjustment system for basic pension funds, and improving the treatment of basic pensions.

**Keywords:** urban and rural residents; basic pension funding; poverty eradication; equalization of basic public services

## 1. Introduction

The sustainable development of a social security system supports the sustainable development of the economy and a harmonious society [1]. The basic endowment insurance system is an important part of the social security system. At present, the sustainability of the basic endowment insurance system is facing severe challenges due to the increase of population aging in China [2].

At present, the basic endowment insurance system in China consists of the basic endowment insurance for urban enterprise employees (BEIUEE), the basic endowment insurance for civil servants and public sector employees (BEICSPEE), and the basic endowment insurance for urban and rural residents (BEIURR). The BEIUEE and BEICSPEE were introduced in the 1950s, mainly for the employees of state-owned enterprises and the staff of government agencies and institutions. The pension system is pay-as-you-go (PAYG). All pensions are undertaken by the state, with the characteristics of obvious planned economy [3]. In 1964, the coverage of social insurance was extended to the employees of companies under collective ownership. The pay-as-you-go system was adopted, but the pension was borne by the companies. In the 1990s, on the one hand, with the deepening of reform and opening-up of the market, the continuous development of market economy, and the continuous expansion of the scale of non-state-owned and collective enterprise employees, the coverage of employee pension insurance necessitated expansion. On the other hand, with the decrease in the fertility rate, the increase of population aging, and the increase of pension pressure, the future solvency of the PAYG system

may presently be at risk under the changing population structure [4]. In this context, the Chinese government has carried out a major reform of BEIUEE. The first measure is to expand the coverage of insurance from state-owned enterprises and collective enterprises to employees and individual workers of all kinds of enterprises in cities and towns. The second is to alter the insurance expenses so that they are shared by enterprises and individuals. Based on the original PAYG system, individual accounts for employees of enterprises are established and implemented in the accumulation system [5]. In 2015, BEICSPEE was also reformed into a partial accumulation system. The BEIURR started in the 1980s, with the aim of resolving the pension problems that occurred after the implementation of the family planning system and household contract responsibility system in rural areas [6]. In 1992, the Ministry of Civil Affairs required that the rural social endowment insurance with "individual payment as the main part and collective subsidy as the auxiliary" should be explored in rural areas where conditions permitted. The complete accumulation system of rural social endowment insurance was quickly cancelled, mainly because, under the traditional Chinese family endowment model, children bear all the funds for the elderly, and the elderly lack the willingness to participate in the full accumulation system of endowment insurance, which requires long-term savings. In addition, the meager income of the rural elderly at that time also restricted their enthusiasm to participate in endowment insurance. After entering the 21st century, in order to narrow the income gap between urban and rural areas, regions and occupations, the Chinese government, drawing lessons from the relatively mature basic endowment insurance for employees, successively launched part of the accumulated endowment insurance system to farmers and urban residents, and in 2014, merged the two into the basic endowment insurance for urban and rural residents [7]. Although the system of partial accumulation is adopted, there are obvious differences between BEIURR, BEIUEE, and BEICSPEE. The social pooling account of BEIURR is not paid by the employer proportionally, but is replaced by the basic pension issued by the government in a unified standard; the individual payment amount of BEIURR does not have to be proportional, but is divided into different grades, and the residents choose the payment amount themselves; in addition, the individual account of BEIURR will be subsidized by the government according to the payment grade in addition to the individual payment.

　　Chinese scholars believe that the problem of sustainable development of BEIURR is more prominent than that of BEIUEE and BEICSPEE. Firstly, although China's BEIURR implements the partial accumulation system, the excessive dependence on government finance results in the "welfare" tendency of BEIURR, which will inevitably seriously affect the sustainable development of the system [8,9]. From the perspective of the development of BEIURR, the tendency of "welfare" will affect the enthusiasm of participating in insurance and payment. BEIURR adopts the method of paying more, making up more, and paying more, encouraging rural and urban residents to participate in social endowment insurance. However, from the perspective of actual payment, due to the fact that government finance bears at least 80% of the payment, the proportion of individual accounts is very small, leading to the majority of the insured choosing the minimum payment standard of 100 yuan per year. This low level of individual payment weakens the use of individual accounts, and the dependence on financial subsidies and basic pensions is more serious [10]. From the perspective of the social security system, the "welfare" of BEIURR not only encroaches on the scope of social assistance, but also weakens the role of social welfare and hinders the development of social assistance and social welfare undertakings. In the long run, it will lead people to hold bad expectations, mistakenly thinking that the state should also be able to bear all the responsibility of personal life, old age, and death. Secondly, the number of insured people of BEIURR is the largest, the aging is the most serious, and the income is the lowest. Data in 2018 show that the number of urban and rural residents participating in basic endowment insurance accounts for 55–56% of the total number of people participating in basic endowment insurance in China, and there are still more than 100 million insured people who are not covered. The Chinese government forecasts that, by 2020, the number of urban and rural residents participating in basic endowment insurance will exceed 600 million [11]. The dependency ratio of BEIURR is 43.56%, which is much higher than that of 16.8% of the whole country. The insured people

of BEIURR are mainly farmers and a few urban residents. In 2018, the per capita disposable income of rural residents was 14,617 yuan, which was far lower than that of urban residents—39,251 yuan [12]. These factors lead to less income and a greater expenditure of BEIURR, which makes it difficult to ensure its long-term sustainability. Finally, the Chinese government plans to build a well-off society by 2020, which requires the goal of getting rid of poverty overall. BEIURR has been given the responsibility of helping the poor through insurance and by increasing the income of the elderly [13,14]. The former requires that the insurance premium be paid by the lowest standard for the poor, while the latter requires that the replacement rate of endowment insurance be increased, which further deepens the "welfare" of BEIURR, reduces the insurance premium rate, and increases the insurance expenditure.

There have been many studies on the sustainability of endowment insurance funds from the perspective of fund revenue and expenditure scale [15–18], profit level [19,20], and investment trend [21–24], etc. Some scholars point out that it is controversial to solve the sustainability problem only by focusing on the contribution/return ratio. If we do not consider growth, wages, effective dependency ratio, and most importantly, asset allocation, we cannot consider sustainability [25–28]. However, BEIURR in China has its own particularity. Firstly, the relevant policies stipulate that "the monthly calculation and payment standard of individual account pensions is currently the total amount of individual account savings divided by 139", which means that the individual account pension does not increase year on year. Secondly, the government's subsidy for individual payment depends on the level of individual payment. The basic pension is determined by the government and is not directly linked to the salary or income of the insured party [29]. Finally, according to the provisions of The Measures for The Administration of Investment in Basic Pension Funds, the proportion of investment in stocks, stock funds, mixed funds, and stock-type pension products shall not be higher than 30% of the net asset value of endowment funds in total; participation in stock index futures and treasury bond future transactions can only be for the purpose of hedging the insured value, and China's pension adopts a sound investment policy—this kind of asset allocation can only ensure the value preservation of fund assets, and cannot cause value-addition. Based on the above reasons, the research on the sustainability of BEIURR should be based on the fund income and expenditure.

The income and expenditure of basic endowment insurance fund is affected by many factors. On the one hand, although more attention has been paid to the changes of population structure and the development of urbanization [30–34], in view of the characteristics of BEIURR, the effective dependency ratio is far higher than the national elderly dependency ratio, so this paper takes the effective dependency ratio as the most important prediction index. On the other hand, as an important part of public expenditure, basic endowment insurance is greatly affected by the policy. This paper takes the change of income and expenditure caused by the policy influence as another important prediction index. As far as we know, our study is the first to investigate the sustainable impact of comprehensive poverty alleviation policy on BEIURR. Some scholars believe that poverty reduction policies may lead to new inequities, worsening rather than improving the structure of income distribution [35–37]. However, our research finds that overall poverty alleviation mainly affects the short-term current deficit, and under the current pension treatment calculation mode stipulated by China, the higher the pension replacement rate, the safer the sustainability of BEIURR in the short term.

## 2. Analysis of the Current Situation of Basic Pension Funds for Urban and Rural Residents in China

To study the sustainability of the basic pension fund for urban and rural residents in China, we must understand its basic status quo. Starting from the development of the basic pension fund for urban and rural residents in China, this paper collates and calculates the current basic information, coverage, dependency ratio, payment rate, and replacement rate of the fund.

After the reform and opening-up, China began to design the basic pension system. According to the requirements of the 7th Five-Year Plan of the People's Republic of China on "speeding up the establishment of rural social insurance system", the Ministry of Civil Affairs of the People's Republic of China (MCA) began to explore the establishment of a rural social pension system as early as 1986.

In 1991, according to the instructions of the State Council, the pilot project was carried out in Mouping, Yantai City, Shandong Province and achieved success. In 1992, on the basis of summing up the pilot experience, the basic scheme of rural social pension at the county level (No.2, 1992, issued by the MCA) was formulated and issued, which was gradually promoted in the areas with conditions in the country, and the rural social pension fund was also established at the same time. In 1999, due to many problems, the State Council issued a notice to rectify and standardize the rural social pension, and asked to stop accepting new business. The audit results of 30 provincial (district, city) rural social pension funds by the Audit Office show that, by the end of 2006, the accumulated income and expenditure of the national rural pension fund were 5.178 billion yuan and 17.139 billion yuan, with a balance of 34.139 billion yuan. In 2009, the state introduced a new rural social pension system. In 2011, the old-age insurance system for urban residents was implemented, pilot projects were selected nationwide for trial implementation, and the old-age insurance fund for urban residents was also simultaneously established. In 2014, the system achieved full coverage in the regions, and a unified basic pension system for urban and rural residents was established. The rural social pension fund and the urban residents' pension fund were merged into the basic pension fund for urban and rural residents. The basic information of the fund since 2014 is shown in Table 1.

**Table 1.** Basic information of the basic pension fund for urban and rural residents (Unit: 1K people, 1billion yuan).

| | Insured Population | Receiving Treatment Population | Income | Expenditure | Accumulated Balance |
|---|---|---|---|---|---|
| 2014 | 501,075 | 143,127 | 231.02 | 157.12 | 384.46 |
| 2015 | 504,722 | 148,003 | 285.46 | 211.67 | 459.23 |
| 2016 | 508,471 | 152,703 | 293.33 | 215.05 | 538.52 |
| 2017 | 512,550 | 155,979 | 330.42 | 237.22 | 631.76 |
| 2018 | 523,917 | 158,981 | 383.77 | 290.55 | 725.03 |

In terms of the dependency ratio of the basic pension,

$$DR = N_r \div N_i \times 100\% = \text{Receiving treatment population} \div \text{Insured population} \times 100\%$$

The dependency ratio from 2014 to 2018 was 39.99%, 41.49%, 42.29%, 43.74%, and 43.56%, respectively, with a significant increase.

From the perspective of insurance coverage, according to the policy, the insured people of the BEIURR are those who have reached the age of 16 (excluding students in school), who are not staff of state organizations and institutions, and urban and rural residents who are not covered by the basic pension system for employees. That is, the number of participants equals the total number minus the sum of the number of students over 16 years old, the number of students under 16 years old, staff of state organizations and institutions, and basic pension participants.

$$N_p = N_t - N_e - N_{u16} - N_n - N_s$$

Through calculation, the number of insured people and the insured rate in 2014–2018 are shown in Table 2.

Data shows that, by the end of 2018, the coverage rate of BEIURR is only 82.6%, and there are still more than 100 million insured people not covered.

In terms of insurance payment rate,

$$\theta = F \div W = \text{payment standard} \div \text{Per capita disposable income}$$

**Table 2.** Number of insured people and the insured rate (Unit: 1million people).

|      | $N_t$ | $N_e$ | $N_{u16}$ | $N_n$ | $N_s$ | $N_p$ | Rate% |
|------|-------|-------|-----------|-------|-------|-------|-------|
| 2014 | 1367.82 | 65.49 | 226.27 | 38.43 | 341.24 | 696.38 | 72.0 |
| 2015 | 1374.62 | 65.26 | 227.99 | 38.56 | 353.61 | 689.19 | 73.2 |
| 2016 | 1382.71 | 65.36 | 230.98 | 38.72 | 379.30 | 668.35 | 76.1 |
| 2017 | 1390.08 | 66.46 | 234.19 | 38.82 | 402.93 | 647.67 | 79.1 |
| 2018 | 1395.38 | 66.93 | 235.89 | 38.95 | 419.02 | 634.60 | 82.6 |

Although the State Council stipulates that the payment standard of BEIURR is set at 12 levels of 100–2000 yuan, according to the data of the 2014–2017 statistical bulletin of human resources and social security development, the individual payment income of the fund increased from 66.6 billion yuan to 81 billion yuan, and the per capita payment increased from 186 yuan to 227 yuan, with an annual growth rate of about 5%. However, due to the differences in the economic development level, the payment rate of residents varies greatly. Take 2018 as an example; the per capita disposable income of residents was 28,228.05 yuan, including 39,250.84 yuan for urban residents and 14,617.03 yuan for rural residents. Therefore, the insurance payment rates of urban and rural residents are as follows:

$$\theta_{c\_2018} \in 0.25\% \sim 5.10\%$$

$$\theta_{r\_2018} \in 0.68\% \sim 13.68\%$$

Not only between urban and rural areas, but also between regions, there is a very significant per capita income gap. Therefore, the State Council stipulates that local governments of all provinces (regions and cities) can add payment grades according to the actual situation. At present, the highest level of payment is 9000 yuan in Beijing. There is a large gap in the minimum payment calibration among different regions, among which Beijing is up to 1000 yuan, followed by Tianjin and Shanghai with 600 yuan and 500 yuan, respectively, and other regions are mostly 100 yuan or 200 yuan. It can be seen that the vast majority of the insured choose the lowest payment level.

From the perspective of insurance treatment, the policy stipulates that the basic pension treatment of urban and rural residents consists of a basic pension and individual account pension, which is paid for life—if the recipient dies, the pension will be stopped from the next month. Individuals who have participated in the BEIURR, who have reached the age of 60, have accumulated payments for 15 years, and have not received the basic pension treatment stipulated by the state, may receive the pension treatment on a monthly basis. The minimum standard of basic pension is determined by the central government. In March 2018, the Ministry of Human Resources and Social Security and the Ministry of Finance were required to establish a normal adjustment mechanism for basic pension and adjust the standard by taking into account the growth of urban and rural residents; income, price changes, basic pension insurance for employees, and other social security standards. In May 2018, the Ministry of Human Resources and Social Security and the Ministry of Finance raised the minimum standard to 88 yuan. At present, the monthly calculation standard of the individual account pension is the total amount of individual account savings divided by 139. The individual account is composed of individual payment, government subsidy, and collective subsidy, and the interest is calculated on an annual basis. At present, there are only two kinds of subsidies stipulated by the central government, no less than 30 yuan below 500 yuan and no less than 60 yuan above 500 yuan. The specific amount shall be determined by the local government. The amount of collective subsidies shall be set by the regions where conditions permit, and there is no unified standard. The formula for calculating the amount of basic pension benefits received by urban and rural residents is:

$$M_{treatment} = M_{basic} + M_{account} = M_{basic} + \left[ \left( M_{individual} + M_{government} + M_{collective} \right) \times \text{Years} + M_{interest} \right] \div \alpha \, (M_0 \text{ is the amount of BEIURR})$$

$M_{basic}$ is basic pension

$M_{account}$ is a personal account pension

$M_{individual}$ is pays for individuals

$M_{government}$ is government subsidy

$M_{collective}$ is the collective subsidy

$M_{interest}$ is interest on the above amount

$\alpha$ is the calculation coefficient

In consideration of the policies applicable to the vast majority of the insured, according to the current regulations of the central government, if $M_{government}$ is less than 500 yuan, it is 30 yuan; if it is more than 500 yuan, it is 60 yuan; since most regions do not have collective subsidy, $M_{collective}$ is not included in the calculation, $\alpha$ is 139, and the interest is 1.5% of the benchmark interest rate of one-year fixed deposit in 2018. The treatment level of BEIURR under different payment grades is shown in Table 3.

**Table 3.** Treatment level under different payment grades (Unit: yuan).

| $M_{individual}$ | $M_{government}$ | $M_{interest}$ | $M_{account}$ | $M_{basic}$ | $M_o$ |
|---|---|---|---|---|---|
| 100.00 | 30.00 | 218.68 | 15.60 | 88.00 | 103.60 |
| 200.00 | 30.00 | 386.89 | 27.60 | 88.00 | 115.60 |
| 300.00 | 30.00 | 555.11 | 39.61 | 88.00 | 127.61 |
| 400.00 | 30.00 | 723.32 | 51.61 | 88.00 | 139.61 |
| 500.00 | 60.00 | 942.00 | 67.21 | 88.00 | 155.21 |
| 600.00 | 60.00 | 1110.21 | 79.21 | 88.00 | 167.21 |
| 700.00 | 60.00 | 1278.42 | 91.21 | 88.00 | 179.21 |
| 800.00 | 60.00 | 1446.64 | 103.21 | 88.00 | 191.21 |
| 900.00 | 60.00 | 1614.85 | 115.21 | 88.00 | 203.21 |
| 1000.00 | 60.00 | 1783.07 | 127.22 | 88.00 | 215.22 |
| 1500.00 | 60.00 | 2624.13 | 187.22 | 88.00 | 275.22 |
| 2000.00 | 60.00 | 3465.20 | 247.23 | 88.00 | 335.23 |

From the perspective of pension replacement rate, taking 2018 data as an example, the national per capita disposable income is 28,228 yuan, including 39,251 yuan for urban residents and 14,617 yuan for rural residents. According to this standard, BEIURR corresponds to the level of pension replacement rate of per capita disposable income of national residents, urban residents, and rural residents. The lowest level is 4.40%, 3.17%, and 8.51%, respectively, and the highest level is 14.25%, 10.25%, and 27.52%, respectively. The replacement rate of BEIURR under different payment grades is shown in Figure 1.

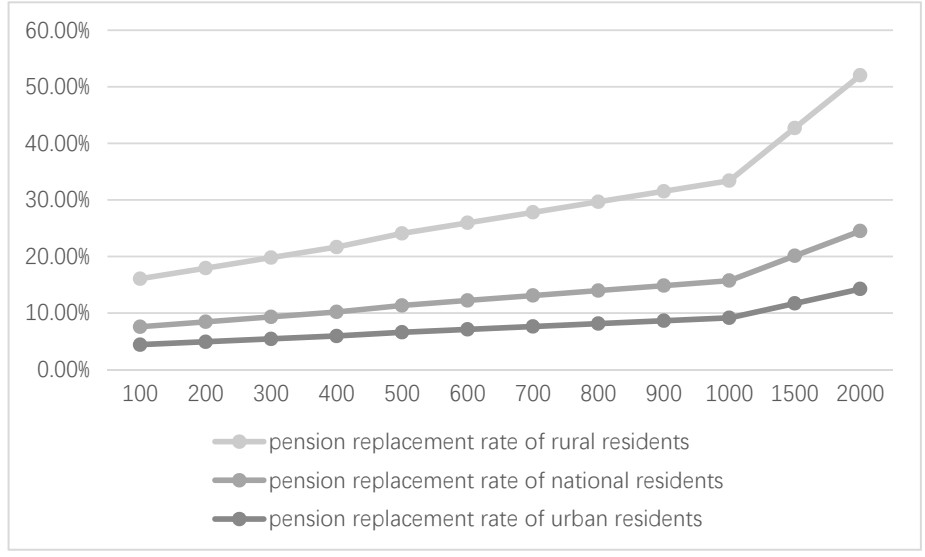

**Figure 1.** Replacement rate of pension in different payment grades.

According to the above data, the current situation of the basic pension fund for urban and rural residents in China can be summarized briefly, as shown in Table 4.

**Table 4.** Current situation of basic pension fund for urban and rural residents in China.

| Project | 2014 | 2015 | 2016 | 2017 | 2018 |
|---|---|---|---|---|---|
| Insured number/10,000 | 50,107.5 | 50,472.2 | 50,847.1 | 51,255 | 52,391.7 |
| Number of people receiving treatment/10,000 | 14,312.7 | 14,800.3 | 15,270.3 | 15,597.9 | 15,898.1 |
| Dependency ratio/% | 39.99 | 41.49 | 42.92 | 43.74 | 43.56 |
| Coverage/% | 71.95 | 73.23 | 76.08 | 79.14 | 82.56 |
| Fund income/100 million yuan | 2310.2 | 2854.6 | 2933.3 | 3304.2 | 3837.7 |
| Fund expenditure/100 million yuan | 1571.2 | 2116.7 | 2150.5 | 2372.2 | 2905.5 |
| Accumulated fund balance/ 100 million yuan | 3844.6 | 4592.3 | 5385.2 | 6317.6 | 7250.3 |
| Payment rate/% | town: 0.35–6.93; countryside: 0.95–19.07 | town: 0.32–6.41; countryside: 0.88–17.51 | town: 0.30–5.95; countryside: 0.81–16.18 | town: 0.27–5.50; countryside: 0.74–14.89 | town: 0.25–5.10; countryside: 0.68–13.68 |
| Pension replacement rate/% | town: 4.31–13.95; countryside: 11.85–38.35 | town: 3.99–12.90; countryside: 10.88–35.22 | town: 3.70–11.97; countryside: 10.06–32.54 | town: 3.42–11.05; countryside: 9.26–29.95 | town: 7–10.25; countryside: 8.51–27.52 |

Source: According to the above analysis and calculation results. Among them, the calculation of payment rate and pension replacement rate is at least 100 yuan at the payment level, and 2000 yuan at most; for the residents' income level, it is divided into urban residents and rural residents; the number of insured and the number of people receiving treatment and other data are from the statistics bulletin of the Ministry of Human Resources and Social Security from 2014 to 2018.

It is found that the financial sustainability of the basic pension fund for urban and rural residents in China is basically stable, but the continuous high dependency ratio and relatively low treatment level also mean that the sustainability of the fund still has defects and deficiencies. From the perspective of the dependency ratio, although the dependency ratio dropped slightly in 2018, reaching 43.56%, it is still significantly higher than the national elderly dependency ratio of 27.80% in the same period. From the perspective of coverage, the current coverage rate of 82.6% still lags far behind the goal of achieving full coverage of pension for urban and rural residents in 2020. From the perspective of insurance payment rate, measured by the 8% payment rate of pension for urban employees in China, the pressure of insurance payment for urban residents is not great, but for rural residents there is a greater pressure of payment in the range of 1500 yuan and 2000 yuan. Limited by the level of disposable income of residents, and with the government incentive effect on different payment grades not being obvious, currently, the insured payment is still at a lower level, with the lowest payment grade as the main level. From the perspective of pension replacement rate, the pension replacement rate of urban residents is too low, which is not attractive to the insured; only the high-end pension replacement rate has a certain attraction to rural residents. From the overall perspective, the pension replacement rate has a significant downward trend, and the basic pension adjustment mechanism needs to be further improved.

## 3. Prediction Model and Related Parameter Assumptions

### 3.1. Sustainable Prediction Model of the Basic Pension Fund for Urban and Rural Residents in China

The purpose of this paper is to study the sustainable development of China's BEIURR fund, that is, to predict its cumulative surplus. Since the forecast is based on the income and expenditure status of the fund, the annual income and expenditure of the fund should be calculated. The forecast model is as follows.

Accumulated fund balance = accumulated fund balance of previous period + fund balance of current period. The basic formula is as follows:

$$F_n = F_{n-1} \times (1 + \beta_1) + F_{current} \times (1 + \beta_2)$$

Formula $F_n$ is the accumulated fund balance in the $n$ year, $\beta_1$ is the average income of the balance of the previous year, and $\beta_2$ is the average income of the current balance. China's pension adopts a sound investment policy, so $\beta_1$ and $\beta_2$ assume a positive value that is very close to zero. For example, the accumulated fund balance in *2020* should be expressed as follows:

$$F_{2020} = F_{2019} \times (1 + \beta_1) + F_{current} \times (1 + \beta_2)$$

Current fund balance equals annual fund income minus annual fund expenditure, expressed as:

$$F_{current} = F_{income} - F_{expenditure}$$

Among them:

Current fund income = number of payers × amount of income = (number of insured − number of people receiving treatment) × (individual payment + government subsidy + collective subsidy); the formula is as follows: Number of people receiving treatment,

$$F_{income} = N_{payers} \times M_{income} = \left(N_{insured} - N_{receiving}\right) \times M_{individual} + M_{government} + M_{collective}$$

Current fund expenditure = number of people receiving treatment × monthly treatment × 12:

$$F_{expenditure} = N_{receiving} \times M_{treatment} \times 12$$

Monthly treatment = basic pension + total amount of individual account ÷ calculation coefficient = basic pension + [(individual payment + government subsidy + collective subsidy) × number of payment years + interest generated by the above amount] ÷ calculation coefficient.

$$M_{treatment} = M_{basic} + M_{account} = M_{basic} + \left[\left(M_{individual} + M_{government} + M_{collective}\right) \times \text{Years} + M_{interest}\right] \div \alpha$$

*3.2. Parameter Assumption*

3.2.1. Number Parameters

Based on the data of the annual population sampling survey of the National Bureau of Statistics of China, this paper uses the cohort component method to predict the total number of people in China in the future. The number of people by age in the current year is equal to the number of people by age in the previous year multiplied by the corresponding survival probability, and the number of new children in the current year is equal to the number of women by age in the current year multiplied by the fertility rate of the corresponding age group. It is estimated that the total population in 2020 will be 1439 million, which is in line with relevant research at home and abroad. In this paper, the prediction result of the medium variant of the population division of the Department of Economic and Social Affairs of the United Nations in 2019, which is closest to the calculation result, is selected as the standard. The population change from 2020 to 2050 is shown in Figure 2.

The population age structure also uses the latest forecast data of the United Nations in 2019. The growth rate of the population under the age of 16 gradually slows down, reaches the peak, and decreases gradually between 2020 and 2050, and the population over the age of 60 gradually increases and accelerates the growth rate.

The number of students aged over 16 is not only related to the population structure, but is also directly related to the school attendance rate. According to the current growth rate of the in school rate, this paper sets that the school rate of colleges and high schools will gradually increase, reaching a peak in 2025 and 2035, respectively—90% for high schools and 70% for colleges and universities—and remain unchanged. The number of civil servants is relatively stable. According to the growth and change of the number of civil servants in China in recent years, the annual growth rate of the number of civil servants in the future is set to be 0.34%.

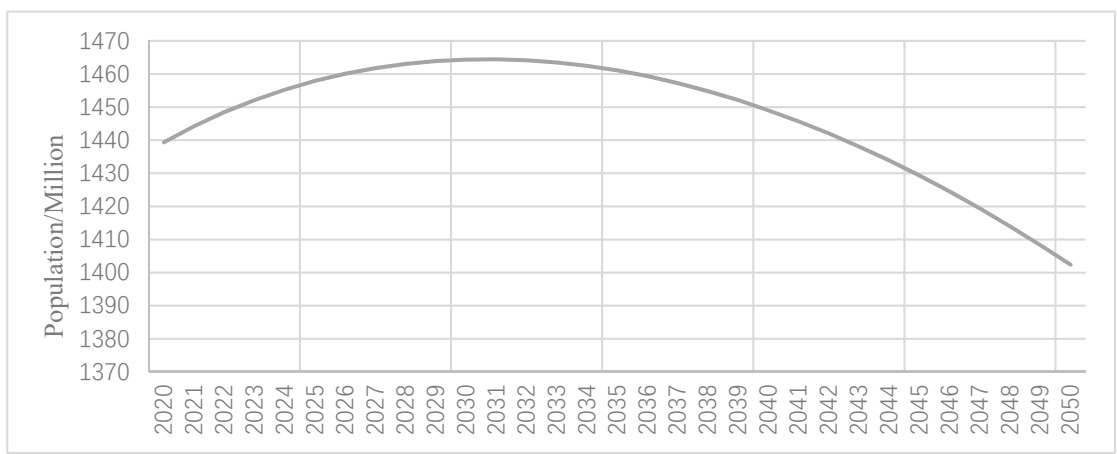

**Figure 2.** United Nations (UN) forecast of China's total population change from 2020–2050.

The number of employees covered by pensions is affected by the number of people in cities and towns. To predict the number of urban and rural residents, we need to consider the development of urbanization, the change of the proportion of urban non-employed people, and the change of population age structure. In order not to fall into the discussion of other problems, but also to simplify the problem reasonably, this paper makes a reasonable assumption of these related parameters. First of all, for the "urbanization rate", the data show that, at the end of 2018, the urbanization rate of the national permanent population is 59.58%. According to the rapid development of China's urbanization at this stage, the relevant departments predict that China's urbanization rate will reach 75% by 2035, so we set the urbanization level according to these predicted results and the current urbanization development level and its growth change. The average growth rate is 1.36%, and will not increase after reaching 80% urbanization rate in 2040. Secondly, for the number of employees participating in the pension, according to the number of insured and the proportion of urban population, the current annual growth rate of this proportion is 2.56%. This paper assumes that the future growth rate is still at this rate, and finally reaches 75% of the employee pension participation rate.

Based on the above relevant analysis and assumptions, we can predict the total population of urban and rural residents, and the number of basic pension target population of urban and rural residents in China from 2016–2050. The results are shown in Table 5.

**Table 5.** Prediction results of basic pension target population and coverage rate of urban and rural residents (Unit: million).

| Year | $N_t$ | $N_{u16}$ | $N_{60}$ | $N_e$ | $N_n$ | $N_s$ | $N_p$ | $N_{insured}$ | Rate% |
|------|-------|-----------|----------|-------|-------|-------|-------|---------------|-------|
| 2014 | 1368 | 240 | 213 | 51 | 38 | 341 | 696 | 501 | 72 |
| 2015 | 1375 | 241 | 222 | 52 | 39 | 354 | 689 | 505 | 73 |
| 2016 | 1383 | 245 | 230 | 53 | 39 | 379 | 668 | | |
| 2017 | 1390 | 247 | 241 | 54 | 39 | 403 | 648 | 513 | 79 |
| 2020 | 1439 | 271 | 250 | 67 | 39 | 467 | 595 | 595 | 100 |
| 2025 | 1458 | 259 | 306 | 81 | 40 | 575 | 503 | 503 | 100 |
| 2030 | 1464 | 248 | 364 | 91 | 41 | 698 | 387 | 387 | 100 |
| 2035 | 1461 | 233 | 393 | 93 | 41 | 822 | 272 | 272 | 100 |
| 2040 | 1449 | 222 | 434 | 90 | 42 | 869 | 226 | 226 | 100 |
| 2045 | 1429 | 216 | 461 | 84 | 43 | 804 | 282 | 282 | 100 |
| 2050 | 1402 | 212 | 485 | 79 | 43 | 841 | 227 | 227 | 100 |

Source: the data from 2014 to 2017 is the actual number. The total population data and the population data of 0–15-year-olds are from the prediction results of the world population outlook: 2019 revision of the population division of the Department of Economic and Social Affairs of the United Nations (UNDESA). Other data are calculated based on the assumption of relevant parameters; the "coverage" is set according to the goal of reaching 100% in 2020.

It is necessary to predict the trend of population aging to determine the number of payers and recipients. According to the data in Table 4, it can be predicted that the proportion of the elderly population in China will be about 17.34% in 2020, 24.86% in 2030, 29.95% in 2040, and 34.59% in 2050. According to this, the number of payers and the number of people receiving treatment can be calculated.

### 3.2.2. Individual Payment Amount

According to the statistics bulletin of human resources and social security development 2014–2017, the individual payment income of the fund increased from 66.6 billion yuan to 81 billion yuan, and the per capita payment increased from 186 yuan to 227 yuan, with an annual increase rate of about 5%. Based on this data, it can be calculated that the per capita payment in 2020 is 262 yuan, 428 yuan in 2030, 697 yuan in 2040, and 1136 yuan in 2050. State subsidies are calculated for the current payment grades of 200 yuan, 400 yuan, 600 yuan, and 1000 yuan, respectively.

### 3.2.3. Basic Pension

According to the guidance of the State Council on the pilot of new rural social pension (GF [2009] No. 32), the basic pension standard of the new rural social pension was 55 yuan per person per month. Since then, the basic pension has been adjusted twice. The first was to increase to 70 yuan in January 2015, and the second was to increase to RMB 88 in January 2018. It is clearly pointed out that the adjustment of other social security standards, such as urban and rural residents' income growth, price change, and basic pension for employees, should be taken into account as a whole to adjust the standards. In fact, from 2015 to 2018, the per capita disposable income has a nominal annual growth rate of about 8.7%. After deducting the price factor, the actual growth rate is about 7.5%, while the annual growth rate of the basic pension is 7.92%. The two are basically consistent. According to the data over the years, China's per capita disposable income growth is relatively stable, so this paper sets the annual growth rate of the basic pension as 7.92%, and adjusts it every three years.

### 3.2.4. Bookkeeping Interest Rate and Market Interest Rate of Individual Accounts

The Guo Fa 2009 No. 32 document and the Guo Fa 2011 No. 18 document both stipulate: "At present, the amount of personal account is calculated by referring to the RMB one-year deposit interest rate of financial institutions published by The People's Bank of China every year." Guo Fa, 2014 No. 8 document stipulates: "The amount of personal account deposit shall be calculated according to the national regulations." Therefore, the bookkeeping interest rate of the individual account of the BEIURR is generally the one-year fixed deposit interest rate issued by The People's Bank of China on 1 January of that year. This paper is based on the current 1.5% setting.

According to the provisions of the measures for the administration of investment in basic pension funds, the proportion of investment in stocks, stock funds, mixed funds, and stock type pension products shall not be higher than 30% of the net asset value of endowment funds in total; participation in stock index futures and treasury bond future transactions can only be for the purpose of hedging the insured value, and China's pension adopts a sound investment policy. In this paper, the market interest rate is 4.08% of the average yield of China's 20-year fixed rate treasury bonds. The above estimates are the reference values of the corresponding parameters.

## 4. Sustainability Prediction of the Basic Pension Fund for Urban and Rural Residents in China

This paper simulates the sustainability of the basic pension fund for urban and rural residents in China without any policy intervention, comprehensive poverty alleviation, or equalization of public services. Based on the prediction of specific data in 2020, the sustainability of the basic pension fund for urban and rural residents in China in the short term (2020–2025) and medium term (2025–2050) is simulated and analyzed. Considering the policy instability, this study does not analyze the long-term sustainability of the fund.

### 4.1. Prediction of Fund Sustainability without Any Policy Intervention

As mentioned above, without any policy intervention, this paper uses the prediction model to bring in relevant parameters to simulate the financial operation of the basic pension fund for urban and rural residents in China. The specific data are shown in Table 6. From the data in Table 6, it can be seen that the basic pension fund income of urban and rural residents presents the fluctuation change of "increase, decrease, increase". In 2020–2025, the income of the fund will increase steadily, because of the continuous increase of the number of insured and the payment amount. However, because of the gradual stagnation of the growth rate of the number of insured, the income growth is not fast. In 2025–2035, the fund revenue will begin to show negative growth, mainly caused by a large reduction in the number of insured. In 2035–2050, the income of the fund will increase again, which is partly due to the substantial increase of the payment amount; more importantly, the short-term fertility tide brought about by the two-child policy implemented in 2011–2015 will cause the total population growth after 2035, and the insured population will increase again. The data show that the growth rate increases from 2040 to 2045, and then the growth rate slows down again. During 2020–2050, the fund's expenditure continues to increase with the deepening of aging and the continuous growth of basic pension.

**Table 6.** Sustainable status of basic pension fund for urban and rural residents (without any policy intervention; Unit: 1 billion yuan).

| Year | $F_{income}$ | $F_{expenditure}$ | $F_{current}$ | $F_n$ |
|------|------|------|------|------|
| 2020 | 143.6 | 156.9 | −13.4 | 779.9 |
| 2025 | 145.1 | 190.7 | −45.6 | 774.6 |
| 2030 | 136.6 | 213.1 | −76.4 | 587.0 |
| 2035 | 120.5 | 234.0 | −113.5 | 169.8 |
| 2040 | 148.5 | 252.2 | −103.7 | −407.5 |
| 2045 | 181.5 | 328.9 | −147.4 | −1203.5 |
| 2050 | 218.3 | 402.5 | −184.3 | −2357.6 |

As the growth rate and absolute value of the fund expenditure are significantly higher than that of the fund income, the fund will begin to run a current deficit in 2020. From 2020 to 2035, the current deficit scale will expand rapidly, and then the growth rate will gradually decrease. However, the current deficit will still increase from 13.4 billion yuan in 2020 to 184.3 billion yuan in 2050. As the fund has a certain accumulated balance in the early stage, it can deal with the current gap of some funds, and the overall number of insured continues to decrease, so the time point when the fund starts to have an accumulated deficit occurs between 2035 and 2040; at that time, the funds of the basic pension for residents will be exhausted and cannot be maintained.

The simulation forecast is based on the situation without any policy intervention, but in reality there will be many policies that affect the sustainable status of the fund. This paper focuses on two policies that will have a great impact on the financial operation of the fund in the short and medium term. One is the policy of "getting rid of poverty in an all-round way", and the other is to realize the equalization of basic public services.

### 4.2. Prediction of Fund Sustainability under the Influence of Helping the Poor Through Insurance

On 27 November, 2015, Xi Jinping pointed out at the central poverty alleviation and development conference that the poor people in rural areas should be lifted out of poverty in 2020. The Ministry of Human Resources and Social Security stressed that the poverty alleviation policy of social insurance should be fully implemented and social insurance for the poor should be fully guaranteed. According to the data, and the rural poverty standard of 2300 yuan per person per year (constant price in 2010), by the end of 2017, there were 30.46 million rural poor people, 12.89 million fewer than at the end of last year (see Figure 3); the incidence of poverty was 3.1%. A large number of poor people join the

basic medical insurance, which will inevitably have an impact on the per capita payment amount, and then reduce the growth expectation of fund income.

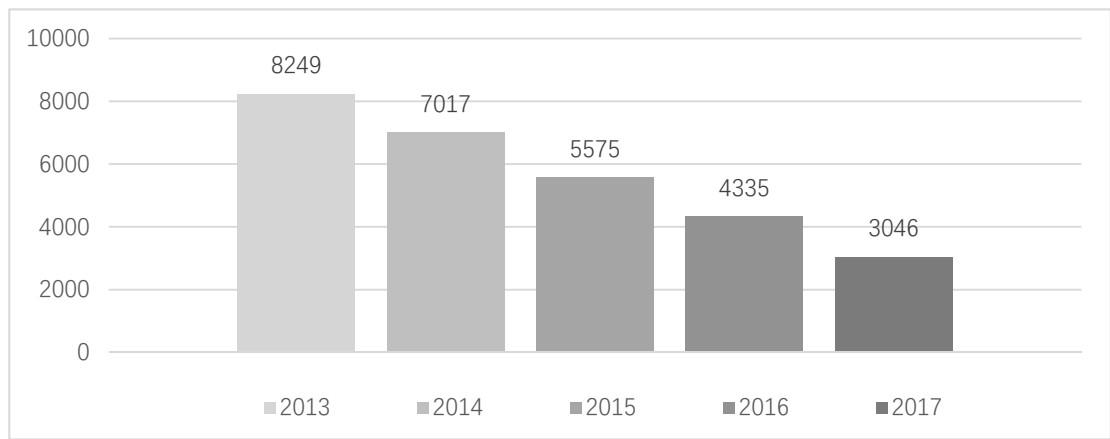

**Figure 3.** Rural poor population in China (10,000 people) for the period 2013–2017.

According to the plan of getting rid of poverty in 2020, this paper assumes that, among the new insured population in 2020 compared with that in 2017, there are 30 million poor people. The number of people who pay and receive treatment is calculated according to the aging standard of that year, and the amount of individual payments is calculated according to the minimum standard of 100 yuan. The specific data are shown in Table 7. Comparing the data in Tables 6 and 7, it can be found that, under the influence of the national poverty alleviation policy, the per capita contribution in 2020 will decrease by about 7% compared with the situation without policy intervention, resulting in the simultaneous decrease of fund income and expenditure. However, due to the larger absolute value of the decrease in fund income, the current deficit has accelerated. In 2020, the current deficit scale is close to 20 billion yuan, up 49% compared with the situation without policy intervention. The early arrival of the current deficit naturally leads to the advance of the accumulated deficit. Under the influence of the comprehensive poverty alleviation policy, the accumulated balance in 2035 will be lower than the current deficit, and the operation of the basic pension fund for residents will be difficult to maintain.

**Table 7.** Sustainable situation of the basic pension fund for urban and rural residents (comprehensive poverty alleviation policy; Unit: 1 billion yuan).

| Year | $F_{income}$ | $F_{expenditure}$ | $F_{current}$ | $F_n$ |
|---|---|---|---|---|
| 2020 | 133.8 | 153.7 | −19.9 | 773.1 |
| 2025 | 135.5 | 186.6 | −51.1 | 734.8 |
| 2030 | 127.4 | 208.2 | −80.8 | 507.1 |
| 2035 | 112.7 | 220.5 | −107.8 | 95.8 |
| 2040 | 138.7 | 245.5 | −106.9 | −540.9 |
| 2045 | 169.1 | 319.5 | −150.4 | −1401.3 |
| 2050 | 203.1 | 389.8 | −186.7 | −2675.6 |

*4.3. Prediction of Fund Sustainability under the Influence of Increasing the Income of the Elderly*

On 18 October, 2017, Xi Jinping put forward two major 15-year struggles in 2020 from 2020 to 2050. From the years 2020–2035, we should basically realize socialist modernization on the basis of building a well-off society in an all-round way, including achieving the equalization of basic public services. As an important part of basic public services, the basic pension must be adjusted according to the policy, which is mainly reflected in the pension replacement rate, which will affect the fund expenditure.

According to the data, the current replacement rate of the basic pension is 4.40–14.25%. With the deepening of population aging and the steady growth of per capita disposable income of urban and

rural residents, people's pursuit of the quality of life of the elderly will continue to improve. Based on the current actual level and development trend of substitution rate, this paper sets the substitution rate of low, medium, and high pension as 3%, 5%, and 10%, respectively, to simulate the sustainable situation of urban and rural basic pension funds. The specific data are shown in Table 8.

**Table 8.** Sustainable status of the basic pension fund for urban and rural residents (equalization of public services; Unit: 1 billion yuan).

| Year | Low Pension Replacement Rate | | | | Medium Pension Replacement Rate | | | | High Pension Replacement Rate | | | |
|------|---------------|-------------------|--------------|-------|---------------|-------------------|--------------|-------|---------------|-------------------|--------------|-------|
| | $F_{income}$ | $F_{expenditure}$ | $F_{current}$ | $F_n$ | $F_{income}$ | $F_{expenditure}$ | $F_{current}$ | $F_n$ | $F_{income}$ | $F_{expenditure}$ | $F_{current}$ | $F_n$ |
| 2020 | 128.9 | 101.6 | 27.3 | 822.2 | 165.9 | 169.4 | −3.5 | 790.2 | 656.1 | 338.7 | 317.4 | 1124.2 |
| 2025 | 133.1 | 152.4 | −19.3 | 947.7 | 270.8 | 254.0 | 16.7 | 987.9 | 870.3 | 508.1 | 362.3 | 1322.3 |
| 2030 | 119.3 | 208.6 | −89.4 | 705.7 | 373.8 | 347.7 | 26.1 | 1207.8 | 1032.6 | 695.4 | 337.2 | 1586.6 |
| 2035 | 180.8 | 226.4 | −45.6 | 636.0 | 437.9 | 377.3 | 60.6 | 1514.1 | 1080.7 | 754.6 | 326.1 | 1925.3 |
| 2040 | 323.9 | 379.9 | −56.0 | 460.6 | 695.2 | 633.1 | 62.1 | 1850.9 | 1623.4 | 1266.2 | 357.2 | 2386.6 |
| 2045 | 531.1 | 603.2 | −72.1 | 156.1 | 1060.5 | 1005.4 | 55.2 | 2252.8 | 2384.0 | 2010.7 | 373.3 | 2933.0 |
| 2050 | 840.6 | 936.8 | −96.2 | −345.8 | 1580.9 | 1561.4 | 19.6 | 2711.3 | 3431.7 | 3122.8 | 308.9 | 3509.4 |

First, the low pension replacement rate scheme is more suitable for China's national conditions in the short term (2020–2025). Under this scheme, the per capita contribution in 2020 is 263 yuan, only 1 yuan higher than the per capita contribution without policy intervention. Compared with the situation without policy intervention, it has the following characteristics. First, in the short term (2020–2025), fund income and expenditure will decrease significantly, but due to the relatively larger proportion and absolute amount of expenditure reduction, the current deficit will be postponed to 2020–2025. The second is that the current deficit presents the phenomenon of "expansion, reduction, and re-expansion". The reduction of the current deficit around 2035 will also benefit from the birth tide formed by the two-child policy. Third, due to the delay in the occurrence of the current deficit and the reduction in its scale, the accumulated balance will not show a deficit until 2050. Therefore, even in accordance with the low level of substitution provided by the current system, the basic pension fund for urban and rural residents will not be able to offset its income by 2025 at the latest; it will reach the first gap peak around 2030 and reach a current deficit of nearly 100 billion yuan in 2050. Due to the continuous and expanding current deficit, the fund will be completely consumed in 2045–2050 and will not be able to continue to operate.

Second, in the medium term (2025–2050), the alternative rate of China's pension will be more suitable for China's actual needs. In the case of the medium pension replacement rate, the per capita contribution in 2020 will be 339 yuan, up 29% compared with the situation without policy intervention, which is difficult to achieve in the short term. However, if the per capita disposable income keeps rising steadily, it will be feasible to reach 600 yuan in 2025. Under this plan, because of the rapid expansion of the insured population in the short term, there will be a temporary and small-scale current deficit. Due to the increase of per capita payment, the current balance will continue to grow, and with the regrowth of the insured population around 2035, it will reach the peak of the current balance. After this, due to the reduction of the number of insured again and the deepening of the aging population, the current balance will decrease, which will be less than 20 billion yuan in 2050, lower than the current balance reduction in 2045–2050. On this basis, we can infer that the accumulated balance of the fund may peak around 2050.

Third, although the scale of the current balance and the accumulated balance of the scheme with the high pension replacement rate are relatively higher, the current balance with large scale can be obtained in the short term under this scheme. The current balance in 2020 will exceed 300 billion yuan, 11 times that of the scheme with the low pension replacement rate, and the peak value of the current balance in the medium term will be up to 373.3 billion yuan. However, it will be difficult for this scheme to meet the needs of China. On the one hand, the rapid increase of fund income in the short term means that the per capita contribution increases significantly, which is hard for residents to

bear. On the other hand, the scale of funds under this scheme is too large, and the current balance will decrease by nearly 70 billion yuan from 2045–2050, and so on. With the aging of population and the decrease of insured population, this scheme is likely to form a huge fund gap at the end of this century, leading China into a "welfare trap".

Through a comprehensive comparative analysis of the three alternatives, we can draw the following conclusions:

First, under the current pension treatment calculation mode stipulated by China, the higher the contribution, the higher the current balance of urban and rural residents' basic pension insurance fund in the short and medium term, and the safer the fund sustainability.

Second, the steady growth of the number of payers plays a decisive role in the sustainability of the fund. The two-child policy implemented in 2011–2015 has greatly improved the financial situation of the fund in the medium-term simulation.

## 5. Conclusions and Suggestions

On the basis of summarizing the current situation of China's basic pension fund for urban and rural residents, this paper established a sustainability prediction model of China. The sustainability of the basic pension fund for urban and rural residents in China in the short term (2020–2025) and medium term (2025–2050) was simulated and predicted. The conclusions are as follows.

First, under the current economic development and existing policy conditions, with the decrease of the total population and the deepening of the aging population, the sustainability of the fund will inevitably encounter challenges.

Second, based on the current calculation method of pension benefits, the per capita payment level will be at a relatively low level for a long time, and the vast majority of the insured and payers will be based on the minimum payment standard. On the one hand, due to the lack of an incentive effect of government subsidies, the gap between subsidies for different levels of payment standards is not large, and on the other hand, because the personal account bookkeeping interest rate is too low.

Thirdly, population parameters, especially the number of contributors, have a decisive impact on the sustainability of the fund. The birth tide brought about by the implementation of the two-child policy from 2011 to 2015 has had a significant positive effect on the medium-term sustainability of the fund.

Fourth, the implementation of the comprehensive poverty alleviation policy in 2020 has no obvious impact on the sustainability of the fund. Overall poverty alleviation in 2020 will mainly affect the short-term current deficit, but not the medium-term cumulative deficit.

Fifthly, comparing different pension substitution rate schemes, under the current pension treatment calculation method stipulated by China, the higher the contribution, the higher the current balance of urban and rural residents' basic pension insurance fund in the short and medium term, and the safer the fund sustainability.

According to the analysis results of the simulation prediction, combined with the actual situation, in order to ensure the sustainable and healthy development of China's basic pension fund for urban and rural residents, the following measures are recommended.

First, improve the population strategy and actively respond to the aging population. First of all, we should adhere to the basic national policy of family planning and encourage birth according to the policy. According to the national population development plan (2016–2030) of the State Council, we should "improve the population development strategy and population policy system, promote the long-term balanced development of the population," adopt incentives, subsidies, and other means, moderately improve the fertility level, and give full play to the effect of the comprehensive two-child policy. We should then strengthen the scientific research on aging and provide intellectual support and theoretical services for the active response to population aging.

Second, improve the adjustment mechanism of BEIURR. The coverage of the BEIURR is "at least 16-year-olds (excluding students in school), non-state organization and public institution staff, and urban and rural residents who are not covered by the basic pension system for employees," that is, rural residents and urban non-working residents. With the deepening of urbanization and the improvement of the basic pension system for employees, the target population of BEIURR are bound to decline significantly. In the simulation forecast, the total population will increase by less than 2% from 2020 to 2030, and the number of the insurance target population will decrease by more than 50%. It is estimated that the target population covered by the basic pension for residents will only account for about 16% of the total population by 2050. At present, these data are 47%. Therefore, it is necessary to clarify the status and role of the basic pension for residents. In the short term (2020–2025), it will be the main pension system with the widest coverage in China. Its main role is to ensure and improve the basic life of the elderly. In the medium term (2025–2050), it will become a powerful supplement to the basic pension for employees, and its main role is to narrow the development gap between urban and rural areas and to ensure the equalization of elderly care services. At present, China has established an initial mechanism for determining the treatment of the BEIURR and normal adjustment of the basic pension. This mechanism should be further supplemented and improved, and an adjustment mechanism for the BEIURR should be established. Government subsidy standards and basic pensions should be flexibly adjusted according to the change of its status and role, in combination with economic and social development and a series of indicators such as payment standard.

Third, we should take comprehensive measures to improve the treatment of the BEIURR. The amount of basic pension benefits received by urban and rural residents is determined by the basic pension and personal account deposits. Personal account deposits mainly rely on individual contributions and government subsidies. Under the premise that the basic pension is uniformly regulated by the central government, the improvement of insurance benefits can be started from two aspects: stimulating the enthusiasm of improving the level of individual contributions and increasing government subsidies. To improve the insurance benefits, we should do the following four works: First, we should improve the government's subsidy and growth rate for individual contributions and establish a government subsidy adjustment mechanism similar to the basic pension adjustment mechanism. The second is to improve the differentiation of government subsidies and step-by-step subsidies of different levels of individual contributions. The third is to increase the interest rate of personal accounts. The accounting interest rate of the basic pension for urban employees has been raised many times in the documents, such as HSTF 2017 No. 71 and HSTF 2017 No. 106, which should be increased accordingly. Finally, it is important to improve the return on investment of the pension fund. The improvement of the return on investment is not only conducive to stimulating the enthusiasm of urban and rural residents to participate in insurance and improve the level of payment, but can also improve their financial situation and ensure the sustainable development of the fund.

**Author Contributions:** Conceptualization, resources, and looking for the link with sustainability, writing—original raft preparation, C.S.; investigation, data curation, L.S. and C.S.; writing—review and editing, L.S., C.S., X.X. All authors have read and agreed to the published version of the manuscript.

**Funding:** National Social Science Fund of China, Grant/Award Number: 15BSH026.

**Conflicts of Interest:** The authors declare no conflict of interest.

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
