# Peer review of "Assessing the Sustainability of China’s Basic Pension Funding for Urban and Rural Residents"

_sustainability, doi:10.3390/su12072833_

Round 1

Reviewer 1 Report

I recommend publication after minor changes.
The references to Sharpe and Blackbseem out of context.

Author Response

Point 1: I recommend publication after minor changes.

Response 1: I have revised the abstract, introduction and part 3.1. MDPI's English editing service is also used to polish the language.

Point 2: The references to Sharpe and Blackbseem out of context.

Response 2: These two references have been deleted. And I rewrote the literature review in the introduction section, which is detailed in lines 32 to 255.

Reviewer 2 Report

In the current format, the standard of English makes the article's logic difficult to decipher. The abstract does not clearly set out the question being explored - and throughout the paper the research question is stated in contradictory ways. The causal direction seems to be testing the impact of a specific policy on the sustainability of the funded rural pension schemes, but this becomes clear only by line 359 - before then it is not clear whether the pensions are part of the policy aspiration being alluded to (poverty alleviation).

The literature review is inadequate, given that it covers a vast amount of ground but in a very superficial way. What is the theoretical and empirical focus of this paper? The literature covers ideas of sustainability, that draw on questions of core ratios between contributions and benefits (which does seem to be the theoretical approach here) but also papers considering broader political economy approaches to asset allocation. These are not explained, and do not align with the rest of the paper, so it is unclear why they are included. It would be better to go into more depth regarding the former literature perhaps.

There is not enough clarity in the section detailing the elements of the current system, and no detail regarded the political process behind the decisions to structure the pension system in the way it is structured. Why was the decision taken to use funded schemes in the first place? Why not use PAYG?

Approaching questions of sustainability by focusing exclusively on contribution / benefit ratios is controversial. The counterargument would be that sustainability cannot be considered without thinking more broadly about growth, wages, the effective dependency ratio, and most importantly, asset allocation. As these issues are not considered, the usefulness of the model, and the persuasiveness of its conclusions, are in question. The authors are free to disagree, but should consider making an argument as to the benefits of their simplistic approach as opposed to a broader perspective.

The topic is however of interest, a significant amount to research material has been gathered. There should be a way to organise it into a paper, taking more care over expression, over linking to the current literature, and evidencing more critical reflection. 

Author Response

Point 1: In the current format, the standard of English makes the article's logic difficult to decipher. The abstract does not clearly set out the question being explored - and throughout the paper the research question is stated in contradictory ways. The causal direction seems to be testing the impact of a specific policy on the sustainability of the funded rural pension schemes, but this becomes clear only by line 359 - before then it is not clear whether the pensions are part of the policy aspiration being alluded to (poverty alleviation).

Response 1: Firstly, I have used MDPI's English editing service to modify the article according to the English specification.

Secondly, I revised the abstract. Make it clear that the research question is "this paper uses actuarial model to evaluate the effect of Poverty Alleviation policy on sustainability of China’s basic pension funding for urban and rural residents (BPFUUR). " See line 238-255 for details.

Thirdly, in the introduction, I have made a credible discussion on Poverty Alleviation policies, “the Chinese government plans to build a well-off society overall by 2020, which requires the goal of getting rid of poverty overall. The basic pension funding for urban and rural residents (BPFUUR) has been given the responsibility of helping the poor through insurance and increasing the income of the elderly”. This paper also forecasts the sustainability of the fund based on these two points, which is detailed in the part 4.2 “Prediction of Fund Sustainability under the Influence of Helping the Poor Through Insurance” and the part 4.3 “Prediction of Fund Sustainability under the Influence of Increasing the Income of the Elderly”, lines1283-1764.

Point 2: The literature review is inadequate, given that it covers a vast amount of ground but in a very superficial way. What is the theoretical and empirical focus of this paper? The literature covers ideas of sustainability, that draw on questions of core ratios between contributions and benefits (which does seem to be the theoretical approach here) but also papers considering broader political economy approaches to asset allocation. These are not explained, and do not align with the rest of the paper, so it is unclear why they are included. It would be better to go into more depth regarding the former literature perhaps.

Response 2: I agree with you that the previous literature review is too superficial. Therefore, the literature review of rewriting focuses on three aspects.

The first is the political process behind the decision-making of the pension system structure and the characteristics of the pension system structure. The second is the possible sustainable problems that these characteristics bring to the pension system. The third is the impact of Poverty Alleviation policy on the sustainability of the pension system.

The details are in lines 32-251.

Point 3: There is not enough clarity in the section detailing the elements of the current system, and no detail regarded the political process behind the decisions to structure the pension system in the way it is structured. Why was the decision taken to use funded schemes in the first place? Why not use PAYG?

Response 3: I must clarify that the purpose of section 2 is to introduce the current situation of basic endowment insurance for urban and rural residents, including the Insured Population, Receiving Treatment Population and so on.

Thanks for your reminding, I realize that the article lacks an introduction to the political process behind the structural decision to the pension system. Therefore, I added a literature review of relevant contents in the part 1 Introduction.

China’s Basic Endowment Insurance for Urban and Rural Residents uses the Partial Accumulation System instead of the PAYG System, which is based on the problems found in the practice of building the Basic Endowment Insurance for Urban Enterprise Employees and the Basic Endowment Insurance for Civil Servants and Public Sector Employees. With the decrease in the fertility rate, the increase of population aging and the increase of pension pressure, the future solvency of the PAYG system may present be at risk under the changing population structure.

The reason for not using the full accumulation system is that there are many problems in the endowment insurance system established in the 1990s, under the traditional Chinese family endowment model, children bear all the funds for the elderly, and the elderly lack the willingness to participate in the full accumulation system of endowment insurance, which requires long-term savings. In addition, the meager income of the rural elderly at that time also restricted their enthusiasm to participate in endowment insurance.

You can see lines 32-161 for details.

Point 4: Approaching questions of sustainability by focusing exclusively on contribution / benefit ratios is controversial. The counterargument would be that sustainability cannot be considered without thinking more broadly about growth, wages, the effective dependency ratio, and most importantly, asset allocation. As these issues are not considered, the usefulness of the model, and the persuasiveness of its conclusions, are in question. The authors are free to disagree, but should consider making an argument as to the benefits of their simplistic approach as opposed to a broader perspective.

Response 4: Thank you very much for your questions. Some scholars have put forward this view. But many Chinese scholars, including me, think that the basic endowment insurance for urban and rural residents in China has its own characteristics.

Firstly, the relevant policies stipulate that "the monthly calculation and payment standard of individual account pension is currently the total amount of individual account savings divided by 139", which means that the individual account pension does not increase year on year.

Secondly, the government's subsidy for individual payment depends on the level of individual payment. The basic pension is determined by the government and is not directly linked to the salary or income of the insured party.

Finally, according to the provisions of The Measures for The Administration of Investment in Basic Pension Funds, the proportion of investment in stocks, stock funds, mixed funds, and stock-type pension products shall not be higher than 30% of the net asset value of endowment funds in total; participation in stock index futures and treasury bond future transactions can only be for the purpose of hedging the insured value, and China’s pension adopts a sound investment policy—this kind of asset allocation can only ensure the value preservation of fund assets, and cannot cause value-added.

Based on the above reasons, this paper forecasts the BEIURR fund sustainability mainly from the fund income and expenditure under the influence of the dependency ratio, so the population parameters are estimated in detail in Section 3.2.1.

You can see lines 891-916 for details.

AUTHORS' SECOND ROUND RESPONSES:

  1. The introduction has improved the sense of the paper.

     Thank you for your recognition of our work.

  1. Non-relevant aspects of the literature review have been removed, but it remains rather superficial, and there seem to be substantial aspects of the paper that are not referenced.

In order to make literature review more in-depth, we add two areas of article citations. One is about the sustainability of the endowment insurance fund, the other is about the factors that affect the income and expenditure of the basic endowment insurance fund.

You can see the details in lines 112 to 144 of the paper.

  1. The sections on method and the conclusion remain coherent and of academic interest.

Thank you for your recognition of our work.

4. The standard of English is not to my judgement at publishable level. There are four errors in the two opening lines (line 11 and 12) alone.

We have modified the relevant content and asked for the help of MDPI English editor again. Hope to meet your requirements.

Reviewer 3 Report

See the report attached

Round 2

Reviewer 2 Report

  1. The introduction has improved the sense of the paper.
  2. Non-relevant aspects of the literature review have been removed, but it remains rather superficial, and there seem to be substantial aspects of the paper that are not referenced.
  3. The sections on method and the conclusion remain coherent and of academic interest.
  4. The standard of English is not to my judgement at publishable level. There are four errors in the two opening lines (line 11 and 12) alone.